# Increasing Acid Tolerance of an Engineered Lactic Acid Bacterium *Pediococcus acidilactici* for L-Lactic Acid Production

**Zhao Yan** , **Mingxing Chen, Jia Jia and Jie Bao** *

State Key Laboratory of Bioreactor Engineering, East China University of Science and Technology, 130 Meilong Road, Shanghai 200237, China; y12190059@mail.ecust.edu.cn (Z.Y.); y45190683@mail.ecust.edu.cn (M.C.); y45180575@mail.ecust.edu.cn (J.J.)
* Correspondence: jbao@ecust.edu.cn; Tel./Fax: +86-21-64251799

**Abstract:** Acid tolerance of the lactic acid bacterium (LAB) is crucially important for the production of free lactic acid as a chemical monomer by simplified purification steps. This study conducts both metabolic modification and adaptive evolution approaches on increasing the acid tolerance of an engineered *Pediococcus acidilactici* strain. The overexpression of the genes encoding lactate dehydrogenase, recombinase, chaperone, glutathione and ATPase did not show the observable changes in acid tolerance. On the other hand, the low pH adaptive evolution showed clear improvement. The L-lactic acid generation and cell viability of the adaptively evolved *P. acidilactici* were doubled at low pH up to 4.0 when wheat straw was used as carbohydrate feedstock. However, the further decrease in pH value close to the pKa (3.86) of lactic acid led to a dramatic reduction in L-lactic acid generation. This result shows a partially successful approach on improving the acid tolerance of the lactic acid bacterium *P. acidilactici*.

**Keywords:** acid tolerance; lactic acid bacterium (LAB); adaptive evolution; metabolic engineering; wheat straw

## 1. Introduction

Industrial lactic acid fermentation by lactic acid bacterium (LAB) is generally at the pH of 5.5~6.5 by adding $Ca(OH)_2$, NaOH or their alkaline compounds forming calcium lactate or sodium lactate [1]. When free lactic acid is required as a chemical monomer for the production of polylactic acid (PLA), sulfuric acid or electrodialysis are used to recover calcium lactate or sodium lactate into free lactic acid; then, this is followed by complicated downstream purification steps [2–5]. To avoid or simplify the complicated purification steps, low pH fermentation is preferred to obtain the free lactic acid directly in a fermentation broth without acidification or without the need for other purification steps. However, the fermentability of lactic acid bacteria is strongly inhibited by the high titer of lactic acid generated with very low pH values [6–8].

Two approaches have been attempted to improve the acid tolerance of lactic acid bacteria: one is metabolic modification [9,10] and the other is adaptive evolution [11,12]. However, only limited results were obtained and far below the demand for industrial applications. In the decarboxylation reaction of amino acids, such as histidine or glutamate, one $H^+$ is consumed, and $CO_2$ is released. The expression of histidine carboxylation in LAB strains enabled the strain to increase intracellular pH and improved acid tolerance [9]. Trehalose, as a protein stabilizer and depressor of nonspecific protein aggregation, played an important role against acid stress [10]. The expression of histidine decarboxylation or the trehalose synthesis pathway only increased the cell survival of *Lactococcus lactis* under low pH with no improvement on lactic acid production [9,10]. Zhang et al. [11] used adaptive evolution to improve the acid tolerance of *Lactococcus lactis*, and the obtained evolved strain showed a slight improvement on lactic acid production, with 17.9 g/L lactic acid at pH 4.3, but more acetic acid was generated.

*Pediococcus acidilactici* is a robust LAB strain with high tolerance to lignocellulose-derived inhibitors and the engineered *P. acidilactici* shows the complete and coordinate xylose utilization of L-lactic acid [13–15]. This study uses both metabolic engineering and adaptive evolution approaches to increase the acid tolerance of *P. acidilactici* ZY271 with xylose utilization using wheat straw as carbohydrate feedstock. The low pH adaptively evolved *P. acidilactici* strain showed the clear improvement in L-lactic acid fermentability at pH 4.0. This result shows the partially successful approach on improving the acid tolerance of the lactic acid bacterium *P. acidilactici*.

## 2. Materials and Methods

### 2.1. Strains and Media

*P. acidilactici* ZY271 and *Lactobacillus acidophilus* La-14 were cultured at 42 °C and 150 rpm in de Man, Rogosa and Sharpe (MRS) medium (20 g/L glucose, 2 g/L dipotassium phosphate, 10 g/L yeast extract, 10 g/L peptone, 5 g/L sodium acetate, 2 g/L ammonium citrate dibasic, 0.58 g/L magnesium sulfate heptahydrate, and 0.25 g/L manganese sulfate monohydrate) [16]. *P. acidilactici* ZY271 was an engineered strain with xylose utilization and stored in the China General Microbiological Culture Collection Center (CGMCC, Beijing, China) with registration number 13611.

*Escherichia coli* XL1-blue were used for plasmid construction and *E. coli* K12 were cultured at 37 °C and 200 rpm in Luria–Bertani (LB) medium (5 g/L yeast extract, 10 g/L peptone, and 10 g/L sodium chloride). To maintain the plasmid stable, 150 μg/mL and 5 μg/mL of erythromycin were separately added into the medium of the *E. coli* and *P. acidilactici* recombinants [15].

*Amorphotheca resinae* ZN1 (CGMCC 7452) used for biodetoxification was isolated in our lab and cultured at 28 °C on a potato dextrose agar (PDA) slant prepared by 200 g of potato juice with addition of 20 g glucose and 15 g of agar in 1 L of deionized water, according to previous studies [17,18].

### 2.2. Reagents and Enzymes

Commercial cellulase Cellic CTec 2.0 was purchased from Novozymes China (Beijing, China). The filter paper activity, cellobiase activity, and protein concentration of the commercial enzyme were 256.0 filter paper activity unit (FPU) per milliliter, 4653.3 cellobiose activity unit (CBU) per milliliter, and 86.3 milligram per milliliter, respectively [19–21].

Yeast extract, peptone, DNA polymerase, restriction endonuclease, and DNA ligase were provided by local suppliers. The reagents dipotassium phosphate, sodium acetate, ammonium citrate dibasic, magnesium sulfate heptahydrate, and manganese sulfate monohydrate were purchased from Lingfeng Chemical Reagent Co., Shanghai, China. L-lactic acid used for adaptive evolution was 88% (*w*/*w*) purity, and was purchased from Titan Scientific Co., Shanghai, China.

### 2.3. Raw Material and Its Biorefinery Processing

The wheat straw was harvested from Nanyang (Henan, China) in the spring of 2018. The pretreatment of the wheat straw was conducted according to our previous study [22]. The pretreated wheat straw contained 33.4 ± 0.1% (*w*/*w*) of cellulose and 3.2 ± 0.2% (*w*/*w*) of hemicellulose according to the protocols [23]. The inhibitors in the pretreated wheat straw included 6.21 ± 0.02 mg/g dry matter (DM) of furfural, 5.00 ± 0.01 mg/g DM of 5-hydroxymethylfurfural (HMF), and 16.65 ± 1.01 mg/g DM of acetic acid. The biodetoxification of the pretreated wheat straw was performed in 15 L helical agitated bioreactor. After biodetoxification, weak acids and furan aldehydes inhibitors were mostly removed [17].

### 2.4. Plasmid Construction

The genomic DNA of *P. acidilactici* ZY271, *E. coli* K12, and *L. acidophilus* La-14 was extracted using a TIANamp bacterial DNA kit (Tiangen Biotech, Beijing, China). The genes

*ldh* with accession number of Gene ID: 29744931 in National Center for Biotechnology Information (NCBI), *ATPase-F0* (Gene ID: 29745541, Gene ID: 29744532, and Gene ID: 29745775), *recA* (Gene ID: 29745849), and *DnaK* (Gene ID: 29744622) were amplified from the genome of *P. acidilactici* ZY271, and the genes *gshAB* (Gene ID: 944881 and Gene ID: 947445) and *ldh3* (Gene ID: 56942534) were amplified from the genome of *E. coli* K12 and *L. acidophilus* La-14, respectively. The constitutive promotor *PldhD* was amplified from the 300 bp upstream sequence from the initiation codon of *ldhD* (Gene ID: 29745642) gene in the genome of *P. acidilactici* ZY271 [15,16]. All the genes were separately inserted into the expression plasmid pZY36e with the promotor *PldhD* at the restriction enzyme's sites of *Pst* I, *Sal* I, or *Xba* I. Six recombinant plasmids, including pZY-*ldh*, pZY-*ldh3*, pZY-*ATPase-F0*, pZY-*recA*, pZY-*DnaK*, and pZY-*gshAB*, were obtained and then transformed into *P. acidilactici* ZY271. All the strains, primers, and plasmids were listed in Table 1.

**Table 1.** Strains, plasmids, and primers used in this study.

| Strains | Characteristic |
|---|---|
| *E. coli* XL1-blue | Host for plasmid construction |
| *P. acidilactici* ZY271 | Parental strain for L-lactic acid fermentation |
| *P. acidilactici* ZY271 (pZY36e) | *P. acidilactici* ZY271 harboring empty plasmid pZY36e |
| *P. acidilactici* ZY271 (pZY36e-*ldh*) | *P. acidilactici* ZY271 harboring expression plasmid pZY36e-*ldh* |
| *P. acidilactici* ZY271 (pZY36e-*ldh3*) | *P. acidilactici* ZY271 harboring expression plasmid pZY36e-*ldh3* |
| *P. acidilactici* ZY271 (pZY36e-*ATPase-F0*) | *P. acidilactici* ZY271 harboring expression plasmid pZY36e-*ATPase-F0* |
| *P. acidilactici* ZY271 (pZY36e-*recA*) | *P. acidilactici* ZY271 harboring expression plasmid pZY36e-*recA* |
| *P. acidilactici* ZY271 (pZY36e-*DnaK*) | *P. acidilactici* ZY271 harboring expression plasmid pZY36e-*DnaK* |
| *P. acidilactici* ZY271 (pZY36e-*gshAB*) | *P. acidilactici* ZY271 harboring expression plasmid pZY36e-*gshAB* |

| Plasmids | Characteristic |
|---|---|
| pZY36e | Expression plasmid by *PldhD* promotor |
| pZY36e-*ldh* | Plasmid for expression of *ldh* by *PldhD* promotor |
| pZY36e-*ldh3* | Plasmid for expression of *ldh* by *PldhD* promotor |
| pZY36e-*ATPase-F0* | Plasmid for expression of *ATPase-F0* by *PldhD* promotor |
| pZY36e-*recA* | Plasmid for expression of *recA* by *PldhD* promotor |
| pZY36e-*DnaK* | Plasmid for expression of *DnaK* by *PldhD* promotor |
| pZY36e-*gshAB* | Plasmid for expression of *gshAB* by *PldhD* promotor |

| Primers | Sequence (5′-3′) |
|---|---|
| *ldh*-F | TGCTCTAGAATGTCTAATATTCAAAATCATCAAAAAGTTGT |
| *ldh*-R | AAAACTGCAGTTATTTGTCTTGTTTTTCAGCAAGAG |
| *ldh3*-F | CTAGTCTAGAATGGCAAGAGTTGAAAAACCTCGT |
| *ldh3*-R | ACGCGTCGACTTATTGACGAACCTTAACGCCA |
| *ATPase-F0*-F | TGCTCTAGAGTGGGTGGTGAATCAATTTCA |
| *ATPase-F0*-R | AAAACTGCAGTCATTTACTCTCACCTAAACCTTCAAT |
| *DnaK*-F | CTAGTCTAGAATGGCAAGTAATAAAATTATTGGTATTGAC |
| *DnaK*-R | ACGCGTCGACTTATTTGTTGTCTTTGTCAGGATCG |
| *gshA*-F | CTAGTCTAGATGCTCTGGTGTGCAGACCAGAC |
| *gshA*-R | ACGCGTCGACTCAGGCGTGTTTTTCCAGCCACA |
| *gshB*-F | ACGCGTCGACATTTGGCGATTTGGGCTAAC |
| *gshB*-R | AACTGCAGTTACTGCTGCTGTAAACGTG |
| *recA*-F | ACGCGTCGACGTGGCAGATGAAAGAAAAGAAG |
| *recA*-R | AAAACTGCAGTTATTTCAAGTCTAATTCAGCTTGGT |

Note: The underline indicates the digestion site.

### 2.5. Adaptive Evolution

The adaptive evolution of *P. acidilactici* ZY271 was conducted in a 100 mL flask containing 20 mL MRS medium with five stages at 42 °C and 150 rpm. The low pH condition was formed by the accumulation of lactic acid generated during the culture. The inoculum ratio of the successive transfer with five stages was 10% ($v/v$). The successive transfers of the Stage I were conducted in MRS medium every 24 h and lasted for 18 transfers. The successive transfers of the Stage II were conducted in MRS medium every 48 h and repeat-

edly performed for 27 transfers. The successive transfers of the Stage III were carried out by an alternate switch between MRS medium and modified MRS medium with initial pH adjusted to 4.0 by 100 g/L L-lactic acid and lasted for 7 transfers. The successive transfers of the Stage IV were performed in MRS medium with 1 g/L sodium acetate every 72 h and repeatedly performed for 39 transfers. The successive transfers of the Stage V were conducted in MRS medium every 48 h and lasted for 109 transfers.

### 2.6. L-Lactic Acid Fermentation

The L-lactic acid fermentability of the recombinants was evaluated in an MRS medium at low pH. Briefly, one colony of the recombinants was inoculated into 5 mL MRS medium in 10 mL tube at 42 °C and 150 rpm for 12 h. A total of 2 mL of the fermentation broth was transferred into 20 mL MRS medium in 100 mL flask for 6 h as seed culture. The seed culture was then inoculated into 20 mL MRS medium in 100 mL flask with 0.3 of initial cell density at the wavelength of 600 nm. The pH value was not controlled during the fermentation, and low pH conditions were formed with the generation of lactic acid.

The L-lactic acid fermentability of the evolved *P. acidilactici* strain was evaluated in an MRS medium and wheat straw at different pH values. The pH values were controlled to 5.5, 5.0, 4.5, and 4.0 by automatic adding 4 M NaOH or 25% (*w/w*) Ca(OH)$_2$ slurry. Briefly, one cryogenic vial of the evolved strain was inoculated into the 20 mL MRS medium in 100 mL flask at 42 °C and 150 rpm for 12 h. A total of 5 mL of the overnight fermentation broth with the same cell density at the wavelength of 600 nm was transferred into 50 mL MRS medium in 250 mL flask for 6 h, and then used as the seed culture for L-lactic acid fermentation. For fermentation in an MRS medium with 90 g/L glucose, the seed culture was directly inoculated into the fermentor containing 500 mL MRS medium at 42 °C and 400 rpm. For simultaneous saccharification and co-fermentation (SSCF), the biodetoxified wheat straw was pre-hydrolyzed at 50 °C and 500 rpm for 12 h with an enzyme dosage of 10 mg total cellulase protein per gram of cellulose, and then the seed culture was inoculated into the 1 L fermentor at 42 °C and 400 rpm [16].

### 2.7. Analysis

Glucose, xylose, and L-lactic acid were analyzed on HPLC (LC-20AD, Shimazu, Kyoto, Japan) equipped with RID-10A detector (Shimadzu, Kyoto, Japan) and a Bio-Rad Aminex HPX-87H column (Bio-Rad, Hercules, CA, USA). A total of 5 mM H$_2$SO$_4$ was used as flow phase at the flow rate of 0.6 mL/min and column temperature was controlled at 65 °C according to our previous study [17].

The cell viability of *P. acidilactici* was assayed at the end of fermentation by counting the colony-forming units (CFU) on the MRS Petri dish when the 100 μL of the $10^{-6}$ or $10^{-7}$ diluted fermentation broth were stretched and cultured for 48 h at 42 °C.

The dosage of neutralizing agents (g/g L-lactic acid) was measured based on the concentration of L-lactic acid and Na$^+$ or Ca$^{2+}$. The concentration of Na$^+$ or Ca$^{2+}$ was detected by ion chromatograph (CIC-D120, Shine, Qingdao, China).

## 3. Results and Discussions

### 3.1. Metabolic Modification of LAB to Improve Acid Tolerance

To increase the acid tolerance of *P. acidilactici* strain, several metabolic modification approaches were conducted by (i) increasing lactic acid pathway flux; (ii) repairing damage by acid stress; and (iii) pumping out the extra intracellular H$^+$ (Figure 1).

Two L-lactate dehydrogenases genes were cloned from the host strain *P. acidilactici* ZY271 (*ldh*) and *Lactobacillus acidophilus* La-14 (*ldh3*) and expressed for the purpose of increasing the flux of lactic acid pathway, but no significant changes were observed.

The genes responsible for DNA or protein damage repair by acid stress were expressed in *P. acidilactici* ZY271. *RecA* encoding recombinase was able to repair DNA damage under low pH conditions [24]; *Dnak* encoding chaperone was reported to renature the impaired protein by acid stress [7]; *gshAB* encoding glutathione synthesis was capable of protecting

LAB cells from acid stress [25]. However, the expression of *recA*, *DnaK*, *gshAB* in *P. acidilactici* ZY271 did not increase acid tolerance.

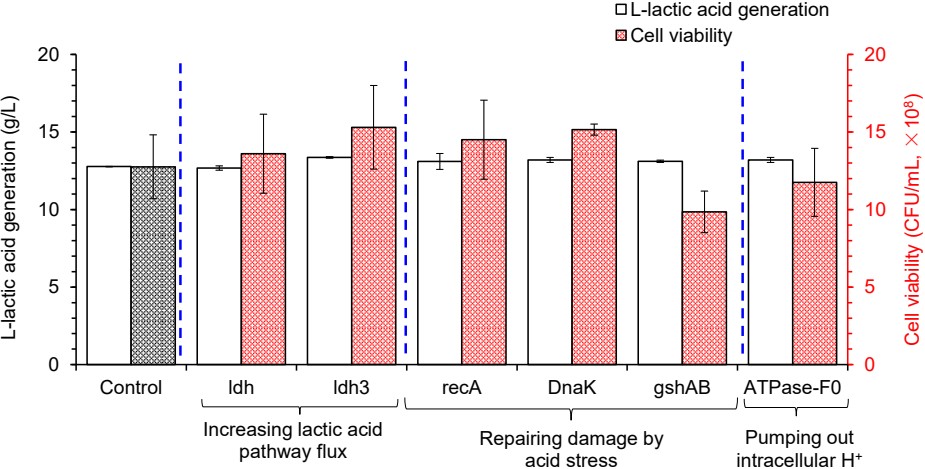

**Figure 1.** L-lactic acid fermentation of *P. acidilactici* recombinants in flasks at low pH. The low pH condition was formed with the generation of lactic acid during fermentation. The *P. acidilactici* recombinant harboring empty plasmid pZY36e was used as the control. The *ldh* and *ldh3* separately encoded L-lactate dehydrogenase from *P. acidilactici* ZY271 and *L. acidophilus* La-14. The *recA* and *DnaK* from *P. acidilactici* ZY271 encoded recombinase and chaperone, respectively. *ATPase-F$_0$* encoded F$_0$ subunit of F$_0$F$_1$-ATPase from *P. acidilactici* ZY271. The *gshAB* encoded glutamate cysteine synthase and glutathione synthetase from *E. coli* K12. The fermentation was performed in de Man, Rogosa and Sharpe (MRS) medium for 24 h, and the condition was control at 42 °C and 150 rpm. Cell viability and L-lactic acid generation were indicated at the end of the fermentation. According to Student's *t* test, the *p*-value of each result was above 0.05.

The gene *ATPase-F$_0$* encoding the F$_0$ subunit of F$_0$F$_1$-ATPase was expressed in *P. acidilactici* ZY271 to pump out the extra intracellular H$^+$. The dissociated H$^+$ from free lactic acid at low pH would impair cell membrane and disturb the biological reactions of LAB [26,27]. The F$_0$ subunit of H$^+$-ATPase was anchored in the cell membrane for pumping out H$^+$, and the number of F$_0$ subunit was limited [26]. F$_1$ subunit of H$^+$-ATPase was dissociative in the cytoplasm for ATP hydrolysis, and the number of F$_1$ subunit was abundant [26]. When intracellular H$^+$ increased, F$_1$ subunit and F$_0$ subunit combined and pumped out H$^+$ with the consumption of ATP [26,27]. Therefore, expressing the F$_0$ subunit of H$^+$-ATPase in *P. acidilactici* ZY271 was conducted for the first time to improve acid tolerance in this study. However, no significant improvement was observed by the overexpression of *ATPase-F$_0$*.

The reasons of the failures of metabolic modifications could come from the inherent property of global response of LAB cells at low pH. Acid stress not only causes lethal DNA or protein damage, but also affects the integrity of cell membrane [8]. The expression of *ldh*, *recA*, *DnaK*, *gshAB*, and *ATPase-F$_0$* may not be sufficient to change the acid tolerance of LAB. The acid tolerance performance by metabolic engineering are more or less affected by certain environmental conditions. The present results did not give the decisive evidences on the function of the genes. We are still working the transporter H$^+$-ATPase for improving the H+ extracellular transportation and lessening the acid tolerance to the cells.

### 3.2. Low pH Adaptive Evolution of LAB and Fermentation Evaluation

Adaptive evolution is a proper method for changing the globe property of cells and has been applied to increase the acid tolerance of LAB [12,28]. This study conducted the long-term adaptive evolution of *P. acidilactici* ZY271 without pH control and a low pH condition was generated by the accumulation of lactic acid (Figure 2). Five stages were switched during long-term adaptive evolution. Stage I was performed in MRS medium until the pH reached 4.0; Stage II was performed by extending the transfer time from 24 h to

48 h until the pH was 3.9; Stage III was conducted with the initial pH adjusted to 4.0 by the addition of extra lactic acid; Stage IV was performed in the medium with the less sodium acetate (from 5 g/L to 1 g/L) to lessen its buffering effect and the pH was allowed to reach 3.6; Stage V returned the culture conditions to Stage II, and the L-lactic acid generation and sugar consumption in this stage (Stage V) fluctuated and then tended to be constant at 3.8. After 200 successive transfers, the lactic acid generation and sugar consumption of *P. acidilactici* strain were almost constant to give a stable adaptively evolved strain.

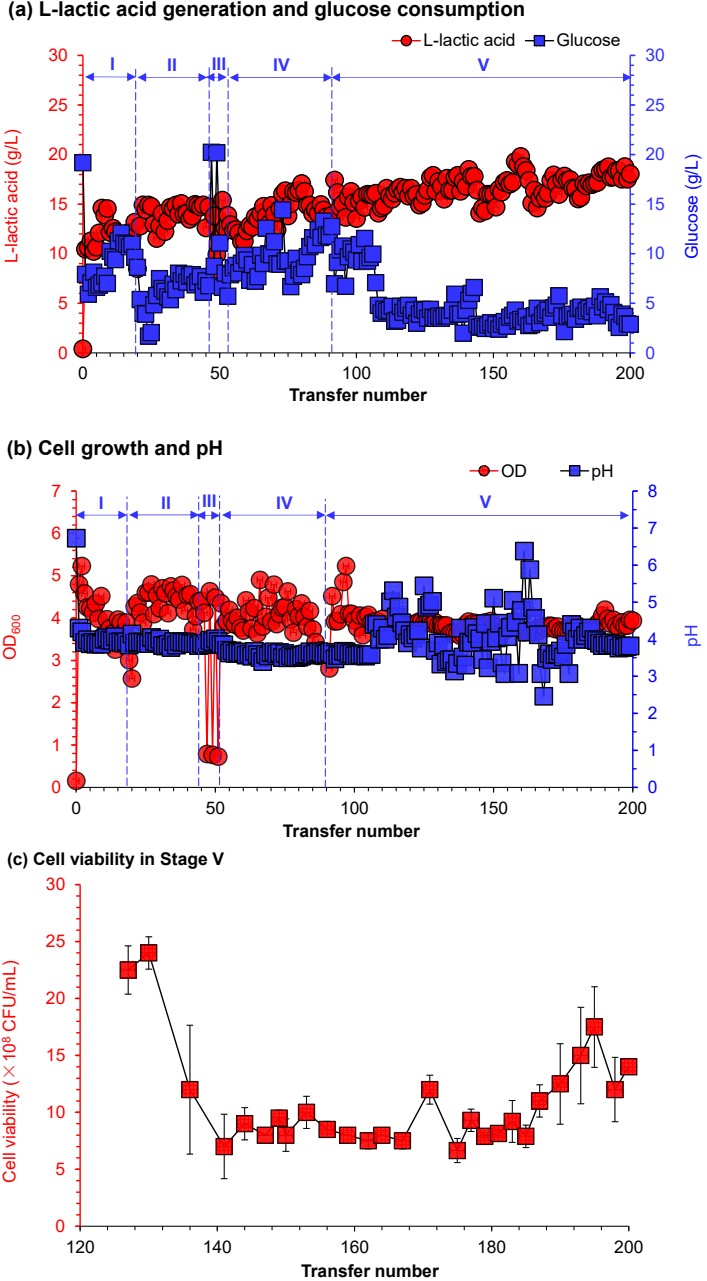

**Figure 2.** Low pH adaptive evolution of the parental strain *P. acidilactici* ZY271. (**a**) L-lactic acid generation and glucose consumption; (**b**) Cell growth and pH; (**c**) Cell viability. Five stages were switched during long-term adaptive evolution. The successive transfers were conducted at 42 °C and 150 rpm, and the inoculum ratio was 10% (*v/v*). $OD_{600}$, pH, L-lactic acid, glucose, and cell viability were measured at the end of each transfer. The details of each stage were in Materials and Methods, Section 2.5.

The evaluation of the adaptively evolved *P. acidilactici* strain was carried out in an MRS medium using pure sugars and wheat straw in simultaneous saccharification and co-fermentation (SSCF). The pH was controlled at 5.5, 5.0, 4.5, and 4.0 by automatic addition of 4 M NaOH with the parental *P. acidilactici* ZY271 as the control (Figure 3). The adaptively evolved *P. acidilactici* strain showed a clear improvement of L-lactic acid generation and cell viability at the lowest pH 4.0 by 34.7% and 4.2-fold using pure sugars (Figure 3a), and 92.3% and 1-fold using wheat straw in SSCF (Figure 3b), respectively, than that of the parental strain. Previous studies also showed that adaptive evolution is helpful for the improvement of the acid tolerance of LAB strains [11,12]. Zhang et al. [11] used adaptive evolution to improve the acid tolerance of *Lactococcus lactis*, and the obtained evolved strain showed slight improvement on lactic acid production with 17.9 g/L lactic acid at pH 4.3. Cubas-Cano et al. [12] used adaptive evolution improved the lactic acid production of *Lactobacillus pentosus* by 6.1 g/L at pH 5.0. In this study, the evolved *P. acidilactici* produced 31.4 g/L L-lactic acid at pH 4.0, 92.3% higher that the parental strain. However, the further reduction in pH value below the pKa of lactic acid (3.86) led to poor cell growth of the adaptively evolved *P. acidilactici* ZY271, indicating the limitation of acid tolerance improvement by adaptive evolution. The lactic acid titer was also lower than that under the regular pH and further efforts should be taken to meet the industrial demand.

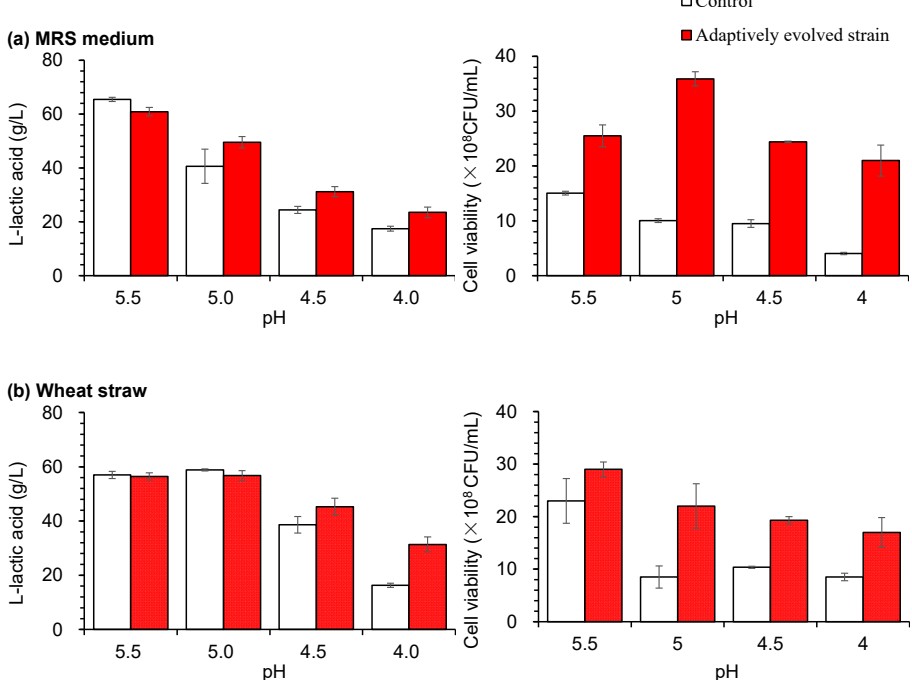

**Figure 3.** L-lactic acid fermentability of the adaptively evolved *P. acidilactici* strain at different pH values in de Man, Rogosa and Sharpe (MRS) medium and simultaneous saccharification and co-fermentation (SSCF). (**a**) MRS medium; (**b**) wheat straw. The parental strain *P. acidilactici* ZY271 was used as the control. The fermentation in MRS medium was conducted in 1 L fermentor containing 90 g/L of glucose for 48 h. The SSCF was performed using biodetoxified wheat straw under 15% (*w/w*) solids loading for 48 h. The condition was controlled at 42 °C and 400 rpm, and the pH was controlled at different values (5.5, 5.0, 4.5, and 4.0) by automatic adding 4 M NaOH. The sugar consumption, L-lactic acid generation, and cell viability were indicated at the end of the fermentation. According to Student's *t* test, the improvement in lactic acid production of the evolved strain was significant, with $p > 0.05$ when the pH value was below 4.5, and the improvement in cell viability of the evolved strain was very significant, with $p < 0.01$ at different pH values.

### 3.3. Neutralizing Agents Reduction for L-Lactic Acid Fermentation under Low pH

NaOH and Ca(OH)$_2$ were used as neutralization agents at pH 4.0 for the production of L-lactic acid from wheat straw using the adaptively evolved *P. acidilactici* strain (Figure 4). When NaOH was used, the NaOH dosage was 0.15 g per gram of L-lactic acid generation to keep the pH at 4.0, equivalent to a 56.3% reduction in NaOH than that at pH 5.5. When Ca(OH)$_2$ was used, the Ca(OH)$_2$ dosage was 0.24 g per gram of L-lactic acid generation, equivalent to a 39.8% reduction in Ca(OH)$_2$ than that at pH 5.5. The results strongly indicated the reduction in the neutralizing agents at low pH by the adaptively evolved strain and the simplified downstream purification steps for free L-lactic acid production [4,29].

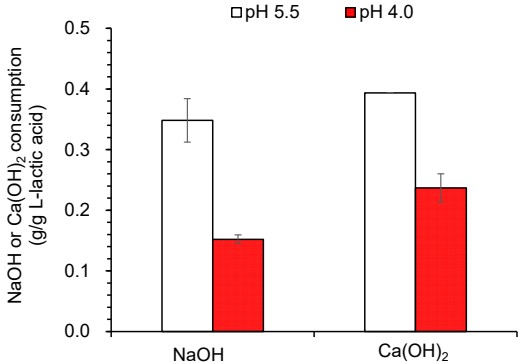

**Figure 4.** The dosage of NaOH or Ca(OH)$_2$ for cellulosic L-lactic acid fermentation in the evolved *P. acidilactici* strain at pH 4.0 and 5.5. The cellulosic L-lactic acid fermentation was performed by simultaneous saccharification and co-fermentation (SSCF) of 5% (*w/w*) solids loading wheat straw for 24 h, and the enzyme dosage was 10 mg protein/g cellulose. The condition was controlled at 42 °C and 400 rpm. The pH was controlled by automatically adding 4 M NaOH or 25% (*w/w*) Ca(OH)$_2$ slurry. According to Student's *t* test, the reduction in NaOH or Ca(OH)$_2$ was significant, with $p < 0.05$.

This study conducted both metabolic modification and low pH adaptive evolution approaches, but only the low pH adaptive evolution in *P. acidilactici* improved the acid tolerance of the engineered *P. acidilactici* strain with L-lactic acid reaching values of 31.4 g/L at pH 4.0. The result reached an obvious progress on L-lactic acid production from lignocellulose feedstock by lactic acid bacterium strains, but is still below the demand of industrial application. More efforts are still required for further breakthroughs in the low pH tolerance of lactic acid bacterium strains.

### 4. Conclusions

Improving the acid tolerance of LAB is crucially important for simplifying the purification steps to obtaining free lactic acid. In this study, the expression of lactate dehydrogenase, recombinase, chaperone, glutathione and ATPase did not increase the acid tolerance of *P. acidilactici*, but the obtained adaptively evolved *P. acidilactici* strain demonstrated improved lactic acid generation and cell viability at low pH during the fermentation of pure sugars and wheat straw. Lactic acid fermentation at low pH could remarkably decrease the dosage of neutralizing agents. This study could provide an efficient approach to improve acid tolerance of LAB.

**Author Contributions:** Conceptualization, supervision, and resources, J.B.; formal analysis: Z.Y.; investigation, Z.Y., M.C. and J.J.; writing—original draft and visualization: J.B., Z.Y., M.C. and J.J. All authors have read and agreed to the published version of the manuscript.

**Funding:** This research was funded by the National Natural Science Foundation of China (31961133006, 21978083).

**Institutional Review Board Statement:** Not applicable.

**Informed Consent Statement:** Not applicable.

**Data Availability Statement:** Not applicable.

**Conflicts of Interest:** The authors declare that they have no known competing financial interest or personal relationship that could have appeared to influence the work reported in this paper.

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
