# Peer review of "Increasing Acid Tolerance of an Engineered Lactic Acid Bacterium Pediococcus acidilactici for L-Lactic Acid Production"

_fermentation, doi:10.3390/fermentation8030096_

Round 1

Reviewer 1 Report

The authors Yan et al. showed a solid work to study how to improve the acid tolerance on the strain P. acidilactici by mainly using two methods: metabolic modification of some relevant genes and adaptive evolution with many passages. And later they proved only the adaptive evolution approach worked in their case. This study brought some points to understand how a LAB strain can adapt to acidic condition after a long term. Ecologically, it is interesting to see that after many generations, how a LAB culture can be evolved to the conditions provided. In the future, I think it will be super interesting to understand mechanistically why this can happen, for example, by checking the changes in their genome content or some protein regulation. In general, personally I like this way to present the results and nicely matched up with the conclusion. I only felt it is a bit weak that the results showed acid tolerance was not improved so much, which was also mentioned in the text. Here are some detailed comments for authors consideration:

last sentence of the abstract:

I don’t understand how the results can be used as a approach to improve acid tolerance for LAB. Also, as I know LAB are really diverse considering their phylogeny and features, I think its not suitable to generalize the conclusion to all LAB strains just based on one strain study. The same for sentences in the intro.

line 81, given full name of the abbreviation for the first time

line 91: maybe it’s better to shortly mention how the straw pretreatment was done, such as if there are still native microbes or not, although you already gave a ref.

line 119: either “alternately switching ” or “an alternate switch”

line 120: Can the authors please show the chemical information of lactic acid you used? Was is pure lac or not?

Do the authors think some co-products like acetate or even ethanol (not sure is the fermentation product) can have some effects on your results? Particularly you used a super high concentration of glucose. Or did your experiments have some controls with the same concentration of acetate but not lactate?

Results 3.3: would you assume the reduction of NaOH and Ca(OH)2 used at lower pH also happen to the non-evolved strain?

I am not so familiar with these gene modifications done for P. acidilactici, but I am wondering if it is generally possible to control the expression of transporters or permeases of LAB strains.  If so, do the authors think there will be some effects regarding the increase of acid tolerance?

Reviewer 2 Report

In this work, the authors describe the optimization of a Pediococcus acidilactici strain for the production of L-lactic acid. Two distinct approaches were used, overexpression of selected genes, endogenous or exogenous, and adaptive laboratory evolution. The latter strategy resulted in an evolved strain with an increased lactic acid titer at pH 4, in particular, when using wheat straw as a substrate.
Overall the manuscript presents interesting data but is somewhat limited since the authors don’t explore the genomic modifications that contributed to this improvement. Thus, it would be interesting to make the whole-genome resequencing of the evolved Pediococcus acidilactici strain and reverse engineering to confirm which genetic modification contributes to this increased lactic acid production.

General comments:
Statistical analysis of the data is missing.
The manuscript should be revised by a proofreader with good written English skills. 

Introduction
Line 45. References reporting the two approaches mentioned here are missing.
Line 46 Please explain what was the aim of the authors when expressing the histidine carboxylation or trehalose synthesis pathway
Line 50. What was the level of lactic acid reached after the adaptive evolution of Lactococcus lactis
Line 53: Please clarify what was engineered and the name of the engineered strain.
It would be important to state what were the production levels for lactic acid reached in other studies, using Pediococcus acidilactici and other relevant organisms.

Materials and Methods
The primers used for the amplification of DNA fragments are missing. 
It would help the reader to include a table listing the organisms used in this work
Line 103: Please include the accession numbers of the amplified genes.
Line 105: The promotor used was 300bp long? Please clarify this. Again, include the accession number of the gene, the primers used for amplification. Please state if this promoter, with the same length, was used in other studies for gene expression, and what type of promoter is this.

Results
Line 162: Please improve the figure legend, state what was the media used, the duration of the fermentation, etc.
Line 166: Include from which organism(s) are the recA and DnaK genes like it was stated for the other genes described in the figure legend.
Line 172: Was this strategy used in previous studies? If so include the references of these studies here.
Line 181: Why was only the Fo subunit from the ATPase expressed? Being an ATPase, the transporter can only pump H+ against a proton gradient if ATP is spent, and this is done by the F1 catalytic subunit. The Fo subunit alone, if functional,  will only work as an H+ channel, promoting proton transport according to the proton gradient, and so the term pump is not correct.
Line 184: The authors have to make the statistical analysis of the data presented throughout the manuscript, and check if the differences observed in the data, are or not statistically significant.

Round 2

Reviewer 2 Report

The authors have improved the manuscript, however there are still some aspects that should be improved before it is suitable for publication.

Question 7: Line 103: Please include the accession numbers of the amplified genes.

Answer 7: All the accession numbers of the amplified genes were added in 2.4. Plasmid construction and marked red.

Comment: The accession numbers presented in the manuscript are not from individual genes, but from contigs that contain multiple genes. The authors should use an identifier specific for each gene. For example GeneID:57366223" that identifies the RecA gene (https://www.ncbi.nlm.nih.gov/nuccore/NZ_CP053421.1?report=genbank&from=1374485&to=1375540&strand=true)

Question 8: Line 105: The promotor used was 300bp long? Please clarify this. Again, include the accession number of the gene, the primers used for amplification. Please state if this promoter, with the same length, was used in other studies for gene expression, and what type of promoter is this.

Answer 8: The promotor PldhDwas defined as 300 bp upstream of geneldhDcontaining the -10 and -35 regions without sequence of the upper gene. The PldhDis a constitutivepromotor and has been widely used in P. acidilacticifor gene expression (Qiu et al., 2020; He et al., 2021). The relevant sentence has been revised in Page 6, Line 112:

“Constitutive promotor PldhDwas amplified from the 300 bp upstream sequence from the initiation codon of ldhD(HMPREF0623_RS01660) gene in the genome of P. acidilactici ZY271.”

References:

Qiu, Z.Y.; Fang, C.; Gao, Q.Q.; Bao, J. A short-chain dehydrogenase plays a key role in cellulosic D-lactic acid fermentability of Pediococcus acidilactici. Bioresour. Technol. 2020, 279, 122473.

He, N.L.; Fang, C.; Qiu, Z.Y.; Bao, J. Increasing sodium lactate production by enhancement of Na+transmembrane transportation in Pediococcus acidilactici. Bioresour. Technol. 2021, 323, 124562.

Comment: The authors should include one of these references in this section

Question 12: Line 181: Why was only the Fo subunit from the ATPase expressed? Being an ATPase, the transporter can only pump H+ against a proton gradient if ATP is spent, and this is done by the F1 catalytic subunit. The Fo subunit alone, if functional, will only work as an H+ channel, promoting proton transport according to the proton gradient, and so the term pump is not correct.

Answer 12: The relevant paragraph has been revised in Page 11, Line 196-201:

“F0subunit of H+-ATPase was anchored in the cell membrane for pumping out H+, and the number of F0subunit was limited. F1subunit of H+-ATPase was dissociative in the cytoplasm for ATP hydrolysis, and the number of F1subunit was abundant. When intracellular H+increased, F1subunit and F0subunit would combine and pump out H+with the consumption of ATP. Therefore, the F0subunit of H+-ATPase was expressed in the P. acidilactici ZY271 to improve acid tolerance.”

Comment: By reading the literature mentioned in this section, it is still not clear why the authors only express one of the H+-ATPase subunits. One of the works mentioned here detected a decreased activity of the H+-ATPase due to a mutation in the catalytic subunit, and did not test the overexpression of the F0 subunit. The other work only measured the activity of the H+-ATPase in different organisms. The authors should clarify if this strategy in particular was used here for the first time, or cite the adequate references.

Question 13: Line 184: The authors have to make the statistical analysis of the data presented throughout the manuscript, and check if the differences observed in the data, are or not statistically significant.

Answer 13: Adaptive evolution could only be one experiment and could not be repeated because the mutations happened in the evolved strain were non-directed. Other experiments were repeated for at least twice and error bars were added on the data.

Comment:The statistical analysis should be done for the data presented in Figures 1. 3 and 4. Please check the statistical analysis done in reference 27, for example.

Author Response

Response to the Reviewers

Reviewer 2: The authors have improved the manuscript, however there are still some aspects that should be improved before it is suitable for publication.

Question 7: Line 103: Please include the accession numbers of the amplified genes.

Answer 7: All the accession numbers of the amplified genes were added in 2.4. Plasmid construction and marked red.

Comment: The accession numbers presented in the manuscript are not from individual genes, but from contigs that contain multiple genes. The authors should use an identifier specific for each gene. For example GeneID:57366223" that identifies the RecA gene (https://www.ncbi.nlm.nih.gov/nuccore/NZ_CP053421.1?report=genbank&from=1374485&to=1375540&strand=true)

Answer: All the accession numbers of the amplified genes were added in 2.4. Plasmid construction. The relevant paragraph has been revised accordingly.

Question 8: Line 105: The promotor used was 300bp long? Please clarify this. Again, include the accession number of the gene, the primers used for amplification. Please state if this promoter, with the same length, was used in other studies for gene expression, and what type of promoter is this.

Answer 8: The promotor PldhDwas defined as 300 bp upstream of gene ldhD containing the -10 and -35 regions without sequence of the upper gene. The PldhDis a constitutivepromotor and has been widely used in P. acidilacticifor gene expression (Qiu et al., 2020; He et al., 2021). The relevant sentence has been revised in Page 6, Line 112:

“Constitutive promotor PldhD was amplified from the 300 bp upstream sequence from the initiation codon of ldhD (HMPREF0623_RS01660) gene in the genome of P. acidilactici ZY271.”

References:

Qiu, Z.Y.; Fang, C.; He, N.L.; Bao, J. An oxidoreductase gene ZMO1116 enhances the p-benzoquinone biodegradation and chiral lactic acid fermentability of Pediococcus acidilactici. J. Biotechnol. 2020, 323, 231-237.

Qiu, Z.Y.; Gao, Q.Q.; Bao, J. Engineering Pediococcus acidilactici with xylose assimilation pathway for high titer cellulosic L-lactic acid fermentation. Bioresour. Technol. 2018, 249, 9-15.

Comment: The authors should include one of these references in this section

Answer 2: The relevant paragraph has been revised accordingly in Page 6 Line 117:

“Constitutive promotor PldhD was amplified from the 300 bp upstream sequence from the initiation codon of ldhD (Gene ID: 29745642) gene in the genome of P. acidilactici ZY271 [15,16].”

Reference:

  1. Qiu, Z.Y.; Fang, C.; He, N.L.; Bao, J. An oxidoreductase gene ZMO1116 enhances the p-benzoquinone biodegradation and chiral lactic acid fermentability of Pediococcus acidilactici. J. Biotechnol. 2020, 323, 231-237.
  2. Qiu, Z.Y.; Gao, Q.Q.; Bao, J. Engineering Pediococcus acidilactici with xylose assimilation pathway for high titer cellulosic L-lactic acid fermentation. Bioresour. Technol. 2018, 249, 9-15.

Question 12: Line 181: Why was only the Fo subunit from the ATPase expressed? Being an ATPase, the transporter can only pump H+ against a proton gradient if ATP is spent, and this is done by the F1 catalytic subunit. The Fo subunit alone, if functional, will only work as an H+ channel, promoting proton transport according to the proton gradient, and so the term pump is not correct.

Answer 12: The relevant paragraph has been revised in Page 11, Line 196-201:

“F0 subunit of H+-ATPase was anchored in the cell membrane for pumping out H+, and the number of F0 subunit was limited. F1 subunit of H+-ATPase was dissociative in the cytoplasm for ATP hydrolysis, and the number of F1 subunit was abundant. When intracellular H+ increased, F1 subunit and F0 subunit would combine and pump out H+ with the consumption of ATP. Therefore, the F0 subunit of H+-ATPase was expressed in the P. acidilactici ZY271 to improve acid tolerance.”

Comment: By reading the literature mentioned in this section, it is still not clear why the authors only express one of the H+-ATPase subunits. One of the works mentioned here detected a decreased activity of the H+-ATPase due to a mutation in the catalytic subunit, and did not test the overexpression of the F0 subunit. The other work only measured the activity of the H+-ATPase in different organisms. The authors should clarify if this strategy in particular was used here for the first time, or cite the adequate references.

Answer: Overexpression of F0 subunit of H+-ATPase was only carried out in this study. The two mentioned literature in this section only stated the importance of H+-ATPase in acid tolerant of LABs. The relevant paragraph has been revised accordingly in Page 11 Line 211:

This sentence “Therefore, the F0 subunit of H+-ATPase was expressed in the P. acidilactici ZY271 to improve acid tolerance.” was revised to “Therefore, expressing F0 subunit of H+-ATPase in P. acidilactici ZY271 was conducted for the first time to improve acid tolerance in this study.”.

Question 13: Line 184: The authors have to make the statistical analysis of the data presented throughout the manuscript, and check if the differences observed in the data, are or not statistically significant.

Answer 13: Adaptive evolution could only be one experiment and could not be repeated because the mutations happened in the evolved strain were non-directed. Other experiments were repeated for at least twice and error bars were added on the data.

Comment: The statistical analysis should be done for the data presented in Figures 1. 3 and 4. Please check the statistical analysis done in reference 27, for example.

Answer: The statistical analysis with T text was conducted in Figures 1. 3 and 4. The relevant paragraph has been revised accordingly in each figure legend:

In Figure 1, this sentence “According to T-text, P value of each result was beyond 0.05.” was added.

In Figure 3, this sentence “According to T-text, the improvement in lactic acid production of the evolved strain was significant with P > 0.05 when pH value was below 4.5, and the improvement in cell viability of the evolved strain was very significant with P < 0.01 at different pH value.” was added.

In Figure 4, this sentence “According to T-text, the reduction of NaOH or Ca(OH)2 was significant with P < 0.05.” was added.